# Optimization of Formulation Parameters in Preparation of *Fructus ligustri lucidi* Dropping Pills by Solid Dispersion Using 2^3^ Full Experimental Design

**DOI:** 10.3390/ph15111433

**Published:** 2022-11-19

**Authors:** Kai-Rong Wu, Wen-Ho Chuo, Yuh-Tyng Huang

**Affiliations:** 1Graduate Institute of Biomedical Science, Chung Hwa University of Medical Technology, Tainan 717302, Taiwan; 2Department of Pharmacy, Tajen University, Pingtung 90741, Taiwan; 3Department of Pharmaceutical Sciences and Technology, Chung Hwa University of Medical Technology, Tainan 717302, Taiwan

**Keywords:** oleanolic acid, *Fructus ligustri lucidi*, water-insoluble, dropping pills, hot-melt method, solid dispersion technology, 2^3^ factorial design, optimal formulation

## Abstract

Oleanolic acid (OA) is an active ingredient of the traditional Chinese medicine (TCM) *Fructus ligustri lucidi* (FLL). Its clinical use is restricted because it is water-insoluble and has limited dosage forms of administration at present. Hence, the FFL dropping pills were prepared by the hot-melt method of solid dispersion technology. A 2^3^ factorial design was used to examine the effects of the materials used to prepare the dropping pills (e.g., different ratios of PEG4000 and PEG6000, FLL extract loading, and percentage of Tween 80) on parameters such as dropping pill roundness, weight variation, and disintegration time. Moreover, 2^3^ full factorial design was utilized to search for the optimal formulation for dissolution experiments. The results showed that the percentage of Tween 80 demonstrated significant effects on dropping pill roundness, weight variation, and disintegration time; FLL extract loading affected roundness and weight variation; and different ratios of PEG4000 and PEG6000 only affected disintegration time. The optimal formulation of the dropping pills released 70% of the drug after 30 min of dissolution release, which was faster than commercially available FLL Chinese medicines. Furthermore, the amount released was higher than that of commercially available formulations. In this study, a solid dispersion technique was used to successfully produce FLL dropping pills. In addition to improving the water insolubility of FLL and increasing the dissolution release percentage of the drug, we increased the application value of FLL and reduced the issues of traditional administration dosage forms.

## 1. Introduction

In traditional Chinese medicine (TCM), *Fructus ligustri lucidi* (FLL) is the dried ripe fruit of *Ligustrum lucidum* from the family Oleaceae. Owing to its rich pharmacological effects, few side effects, and low toxicity, it is widely used to treat hepatitis, diabetes, hyperlipidemia, atherosclerosis, menopausal syndrome, infertility, and tumors in clinical practice [1,2,3,4]. However, FLL is currently administered as a Chinese medicine powder or as traditional decoctions. The active ingredient with the highest FLL content is oleanolic acid (OA), a triterpenoid compound that is practically insoluble in water (1.748 μg/L). [5,6]. In addition, the traditional decoction is prepared by decocting with water, which is not conducive to the extraction and dissolution of OA; thus, this method of administration will affect the absorption of the OA in the body, resulting in poor bioavailability that affects its efficacy, which limits the clinical use of FLL. To increase the efficacy of FLL, many previous studies have employed traditional compatible methods, traditional Chinese medicine concoctions, and nano-pulverization to increase the OA content and increase its efficacy [7,8,9,10,11,12]. However, there are numerous drawbacks of the FLL Chinese medicine powder in clinical settings, such as the bad taste of the medicine when taken orally, the inconvenience of carrying the formulation, and the hygroscopic nature of the powder to environmental moisture [13]. Due to the presence of these issues from the dosage forms of Chinese medicine powder and traditional decoction at present, the purpose of this study was to design a simple, convenient, and fast solid dispersion technique for the preparation of dropping pills to overcome the current situation.

The solid dispersion (SD) technique is a method that uniformly disperses water-insoluble drugs in a hydrophilic inert solid carrier in molecular, colloidal, microcrystalline, or amorphous forms to increase the solubility and dissolution rate of the poorly soluble drug, thereby improving its bioavailability [14,15,16]. Dropping pills are currently the most common dosage form for herbal medicines prepared using the SD technique. TCM dropping pills are prepared by heating and melting TCM extracts and hydrophilic carriers (matrix or excipients) to form a suspension, then dropping the suspension into an immiscible condensate in the form of drops, and the surface tension of the condensate causes the drops of the suspension mix to shrink into a circular shape and condense into a solid drop form [17]. It has many advantages, such as simple preparation, accurate dosing, stable quality, high bioavailability, fast absorption, and portability [18,19]. The authors of previous literature have conducted in-depth studies on preparation methods, the development of sustained-release or immediate-release products and clinical trials of dropping pills [20,21]. However, very few studies have examined the effects of the amount and type of materials, and TCM extracts used in dropping pill preparation on the appearance, weight variation, and disintegration of dropping pills.

Experimental design is used to design experiments by screening the experimental conditions, decreasing the effects of extrinsic factors on the experiment. Data can be collected; correlation between factors and problems can be identified in fewer experiments with lower experimental costs and time through efficient methods. This method has a broad application range. A factorial experimental design examines the effects of multiple factors. In this design, combinations of all the possible factor levels are tested, and the experimental sequence is random. The experimental results can be used to examine the effects of the main response factor and multi-factor interactions on the response [22,23,24].

Therefore, the SD technique was used to prepare dropping pills to improve the drawbacks of traditional FLL dosage administration forms and overcome the water insolubility of the OA. A 2^3^ factorial experimental design was used to examine the effects of the materials used for dropping pill preparation on the roundness, weight variation, and disintegration time. Moreover, it was further used to screen an optimal formulation for comparison with the commercially available Chinese medicines in an in vitro dissolution test. We expected that an optimal dropping pill formulation could increase the solubility and release % of OA in FLL, enhance the in vivo efficacy of the drug, and improve the issues of traditional dosage administration forms. In the future, the optimal dropping pill formulation can be provided to pharmaceutical companies as a reference to scale-up production.

## 2. Results and Discussion 

### 2.1. FLL Concoction

Figure 1 shows FLL before and after concoction using Shaoxing wine. The appearance of the FLL product after wine steaming was brownish-black with a white powder on its surface. Due to the concoction process, the preparation of the FLL resulted in a loose texture, increased permeability, with increased diffusion and release of oleanolic acid, resulting in the white powdery appearance of the FLL [25].

### 2.2. Preparation of FLL Dropping Pills

#### 2.2.1. Preliminary Test (Variables and Level Range Screening)

##### Type and Ratio of Substrate

Screening of the type and ratio of the substrate was performed with FLL extract fixed at 20% and 5% Tween 80. The results showed that when polyethylene glycol (PEG) 4000 was used as a substrate, the drug and substrate melted rapidly, whereas the droplets were loose and soft, and easy to drop. However, the dropping pills were poorly formed, unevenly round, and softer. When PEG 6000 was used as a substrate, owing to its high viscosity, the drug and substrate melted more slowly and over a longer time, and the droplets were viscous and dropped slowly. Although the dropping pills were slightly round, their size uniformity was poor and hardness was higher. We deduced that the reason for this was that the appearance and viscosity of the PEG product was affected by the molecular weight. A higher PEG molecular weight led to a more solid product. In contrast, when the molecular weight was lower, the product was in a liquid or semi-solid state [26]. Therefore, in the preliminary test, the prepared dropping pills were well-formed and the dropping pills were round when a specific ratio of PEG4000 and PEG6000 was mixed and used (Figure 2). Hence, the type and ratio of substrate was set at a PEG 4000:PEG 6000 ratio of 3:1 and 1:3 for the further evaluations of the 2^3^ factorial experimental design. 

##### Condensate Selection

The substrate ratio and type were fixed as 1:1 PEG 4000:PEG 6000, the FLL extract loading was 20%, and Tween 80 was used at 5% to screen the condensate using liquid paraffin, polydimethylsiloxane 100, and polydimethylsiloxane 350. The surface tension and viscosity of the condensate must be considered in condensate selection as this will affect the sedimentation velocity of the drops. The experimental results in Figure 3 showed that when liquid paraffin was used as the condensate, the surface tension was larger (35 mN/m), the viscosity (>37 mm^2^/s) was lower [26], and the droplet settled faster. Hence, the droplet became spherical in the condensate due to surface tension. During rapid sedimentation, the droplet gradually became flat due to gravity acting on the droplet. At the base of the condensation column, the dropping pill was almost oblate. When polydimethylsiloxane 350 was used as the condensate, as its viscosity (332.5–367.5 mm^2^/s) and specific gravity are larger, the droplet first stopped when it was dripped on the condensate surface before settling slowly. During sedimentation, the droplet gradually became spherical due to the effects of surface tension. As polydimethylsiloxane 350 is viscous, gravity acting downwards on the droplet and viscous friction force acting upwards from the condensate causes the spherical droplet to gradually elongate. Therefore, the prepared dropping pills were elongated and their roundness was poor. In addition, polydimethylsiloxane 350 is viscous and is difficult to clean after use. When polydimethylsiloxane 100 was used as the condensate, its surface tension (20.5–21.2 mN/m) and viscosity (95–105 mm^2^/s) were moderate [26]. Hence, it provided a suitable sedimentation velocity and resulted in a good dropping pill shape. Therefore, we used polydimethylsiloxane 100 as the condensate in the subsequent 2^3^ factorial experimental design for preparing dropping pills.

##### FLL Extract Loading

The effects of adding different ratios of FLL extract (20, 25, 30, 35, and 40%) to the formulation on the dropping pill state and shape were assessed when the PEG 4000:PEG 6000 ratio was 1:1, the Tween 80 was used at 5%, and polydimethylsiloxane 100 was used as the condensate. As shown in Figure 4, the droplet viscosity was suitable and a good pill shape was formed when the FLL extract content was 20–35%. When the FLL extract loading was 40%, the droplet blocked the nozzle, as it was too viscous, and preparation could not be carried out. As the ratio of FLL extract in the formulation affects its viscosity, an overly high extract ratio would lead to very viscous droplets and difficulty in formation, resulting in poor dropping pill form, and may even result in the inability to carry out dropping pill preparation. When the FLL extract loading is low and the substrate ratio is high, its melting and dispersion are better. Although this can somewhat improve the dropping pill roundness, the drug extract content is relatively low. Meanwhile, this will increase the administration dose. Therefore, after considering the dropping pill roundness and drug loading, we set the FLL extract level at 20–35% in the 2^3^ factorial experimental design.

##### Surfactant Selection

Surfactant screening was carried out with the substrate fixed as PEG 4000:PEG 6000:1:1, 20% FLL extract, and polydimethylsiloxane 100 as the condensate. The experimental results demonstrated that dropping pills without surfactant were slower to melt, had more viscous droplets, and had the slowest dripping speed compared with other groups containing surfactant. In formulations with surfactants added, the melting time of the FLL extract and the surfactant was shortened, and the dripping speed was faster. Additionally, in the groups containing surfactant, the molten state of FLL extract and substrate, the droplet viscosity and speed were better in the Tween 80 groups than in the Span 80 and Span 60 groups. Furthermore, when disintegration experiments were conducted for dropping pills with Tween 80, we found that the disintegration time was shorter than for dropping pills containing Span 80 and Span 60. The results showed that using Tween 80 significantly improved the solubility of the FLL extract in the substrate. These solubilization results were the best among the three surfactants. We speculate that this is because the HLB of Tween 80 is 15, which is within the HLB range for solubilizers, whereas that of the other two surfactants is not [26,27]. Therefore, we selected Tween 80 as the variable for 2^3^ experimental design evaluation, and the range was set at 5–15% (Figure 5).

### 2.3. The 2^3^ Experimental Design

In this study, the 2^3^ full factorial design was used to estimate three independent variables in the formulations of dropping pills including the ratio of PEG4000/PEG6000 (A), FLL extract loading (B) and percentage of Tween 80 (C); the dependent variables selected were roundness, weight variation (%) and disintegration time, to screen for the optimal formulation. The different formulations of the factorial design consisted of all possible combinations of all factors at all levels and were conducted in a fully randomized order. The matrix of the experiments and results of the responses for every experiment are listed in Table 1. The statistical evaluation of the results was carried out by analysis of variance (ANOVA) using a commercially available statistical software package (DESIGN EXPERT V 6.0.3, Minneapolis). The results in Table 1 showed that the roundness range was 0.68–0.96, the weight variation range was 3.85–8.14%, and the disintegration time range was 4.06–6.94 min for dropping pills prepared using the eight formulations listed in the 2^3^ factorial experimental design. The results from ANOVA showed that FLL extract loading (B) and percentage of Tween 80 (C), and the interaction factors between the ratio of PEG 4000/PEG 6000 (A) and FLL extract loading (B) significantly affected roundness; extract loading (B) and percentage of Tween 80 (C) significantly affected weight variation; and the ratio of PEG 4000/PEG 6000 (A) and percentage of Tween 80 (C) significantly affected disintegration time

Figure 6 provides a half-normal probability plot, which is the cumulative normal probability of the response values and their relationships. The results of Figure 6 showed the important factors affecting dropping pill roundness, weight variation, and disintegration time. If a factor is along the line, its effect on the dependent variable can be ignored, but factors with significant effects on dependent variables are far away from the line. Figure 6a shows that FLL extract loading (B), percentage of Tween 80 (C), and the interaction factors between ratio of PEG 4000/PEG 6000 (A) and FLL extract loading (B) were further from the line; Figure 6b shows that FLL extract loading (B) and percentage of Tween 80 (C) were further from the line, as well as Figure 6c shows that the ratio of PEG 4000/PEG 6000 (A) and percentage of Tween 80 (C) were further from the line. Factors that were further from the line in Figure 6a–c had statistically significant effects on the dependent variable. These results are consistent with those observed in Table 1 from the ANOVA analysis [28].

Figure 7a,b show the correlation between the roundness and factors B and C which have significant effects supported by the ANOVA analysis, respectively. The figures demonstrate that the higher the FLL extract loading (B), the lower the roundness, and vice versa; the higher the concentration of Tween 80 (C) added to the formulation, the lower the roundness. Figure 7a shows that the roundness decreased from 0.96 to 0.72 when the FLL extract loading increased from 20 to 35%. In Figure 7b, the roundness decreased from 0.96 to 0.85 when the Tween content increased from 5 to 15%. Therefore, we found that FLL extract loading (B) and percentage of Tween 80 (C) negatively affected roundness. This may be because an increase in FLL extract loading increases the droplet viscosity when adding to the condensate and makes it difficult to form a round shape. Sometimes, tailing may occur. Additionally, as the Tween 80 concentration increased, its solubilization increased the melting speed of the FLL extract and substrate, resulting in loose and soft droplets and a faster dropping speed. This caused the dropping pills to not be round or uniform in size, which is consistent with previous studies [26,27].

In addition to the evaluation of single variables, interactions are also key to obtaining optimal conditions. From the statistical results of the experimental design, we found that the ratio of PEG 4000/PEG 6000 (A) and FLL extract loading (B) jointly affected roundness, as shown in Figure 8a,b. When the PEG6000 content in the formulation was increased (high level), the effects of FLL extract loading on roundness were more significant than during a low PEG6000 content (low level). When the PEG4000:PEG6000 ratio was 1:3, the higher the FLL extract loading, the lower the roundness. This is because PEG600 is a solid and hard substance, which increases the viscosity when added to a liquid. The droplets become too viscous and are unable to form a round shape, resulting in decreased roundness. If the FLL extract loading was 20%, increasing the ratio of PEG6000 made the dropping pills more round (increased roundness). This may be because PEG is a plasticizer, and its molecular weight makes it a liquid, semi-solid, or solid. Therefore, adding different molecular weights of PEG and the amount of PEG added affect the appearance of the dropping pills formed. Hence, it is recommended that proportional adjustments can be made to the formulation [26]. Figure 8 also showed that the concentration of Tween 80 added also affected roundness, but the magnitude of the effect was related to the type of PEG added. When the PEG6000 content was higher, the roundness was affected regardless of FLL extract loading; when the PEG4000 content was high, the effects of Tween 80 concentration on roundness were not great, regardless of FLL extract content; e.g., when the PEG 4000:PEG 6000 ratio was 3:1, 15% Tween 80 was used, and the FLL extract loading was 20%, the roundness was 0.79. Increasing the FLL extract loading to 35% resulted in a roundness of 0.80, and the difference between the two was small.

The ANOVA analysis showed that FLL extract loading (B) and percentage of Tween 80 (C) significantly affected weight variation. The results are shown in Figure 9a,b, in which FLL extract loading (B) and percentage of Tween 80 (C) positively affected weight variation. This meant that increasing FLL extract loading and Tween 80 concentration would increase the weight variation of dropping pills. Figure 9a showed that when the FLL extract loading increased from 20 to 35%, the weight variation increased from 5.30 to 6.73. The same trend was observed in Figure 9b, where increasing the Tween 80 concentration from 5 to 15% increased the weight variation from 4.50 to 7.50. This may be because an increase in the Tween 80 concentration causes an increase in the melting speed of the FLL extract and substrate, resulting in low viscosity, soft droplets, and a fast dropping time. The dropping pills formed were not round and not uniform in size, resulting in increased weight variation [26].

Figure 10a,b shows the correlation plots of the ratio of PEG 4000/PEG 6000 (A) and percentage of Tween 80 (C) with significant effects on the disintegration time. Figure 10a shows that increasing the PEG 6000 ratio increased disintegration time. Conversely, when the PEG4000 content was higher than the PEG6000 content, the disintegration time was shorter. We speculate that this is because PEG6000 has a higher molecular weight and hardness, and increasing its ratio may delay dropping pill disintegration [26]. Figure 10b shows the effects of the percentage of Tween 80 on disintegration time. When the percentage of Tween 80 was increased from 5 to 15%, the disintegration time was significantly shortened from 6.30 to 4.29 min. This is because Tween 80 is a non-ionic surfactant, and an increase in its concentration increases the contact area between the dropping pill and the disintegration liquid, which accelerates the infiltration of liquid into the dropping pill, causing disintegration and drug release [26,27].

### 2.4. Evaluation of the Optimal Formulation

The ultimate goal of this experiment was to achieve uniformly sized dropping pills with a near-round shape and to increase dissolution rate by disintegration in a short period of time; hence, we referred to previous literature for screening the optimal formulation [19,29]. The optimal conditions for dropping pills were set as a roundness close to 1, a weight variation <5%, and a disintegration time of 5–15 min for screening in this study. Statistical software and experimental design were used to deduce the formulation to achieve the aforementioned optimal conditions. From the results, we found that the dropping pill could meet these criteria when the ratio of PEG 4000/PEG 6000 (A) was set as 1:3, FLL extract loading (B) was 20%, and percentage of Tween 80 (C) was 6.45%. The optimal formulation dropping pill (Figure 11) prepared using these conditions was then evaluated. The results showed that the roundness was 0.95, the weight variation was 4.21%, and the disintegration time was 6.27 min, which met the optimal conditions.

To ensure the uniformity of the OA content in the optimal dropping pill formulations, content uniformity was determined. Table 2 shows the results of the content uniformity test. From the experimental data, it is evident that the content of OA in the 10 groups of the optimal dropping pills ranged from 86.29% to 96.45%, with an average value of 91.54% and a C.V. of 3.40%, which is in accordance with the specification of content uniformity [17]. In addition, from the results of the content uniformity test, it was found that in the process of making the optimal dropping pills using the solid dispersion technique, the FLL extract was well mixed with the substrate by the hot-melt method and the FLL extract was evenly dispersed in the substrate. 

The data from the solubility test in Table 3 showed that this study indeed improved the solubility of OA by using of the solid dispersion technique to prepare the dropping pills.

The results in Figure 12 show the dissolution curves of the optimal formulation of the FLL dropping pill and commercial Chinese patent medicines. The optimal formulation dropping pills started releasing the OA after 5 min of dissolution, and 70% of the OA was released at 30 min. After 30 min, the release rate decreased, and around 80% was released at 2 h. The release % of the commercial Chinese patent medicines was slower than the optimal formulation.

In this study, the results of the solubility and dissolution tests were consistent with previously published literature on the use of solid dispersion techniques to improve the dissolution rate and solubility of water-insoluble drugs [14,15,16]. It is speculated that the water-insoluble drug easily forms an amorphous compound in the hydrophilic carrier resulting in an increase in the surface area and the improvement of the wettability of the compound; hence, the dissolution rate and solubility of the water-insoluble drug can be enhanced [30].

Finally, from all the results of this experiment, we found that using a solid dispersion technique to make a dropping pill for a water-insoluble drug can not only increase the release rate, but also improve the solubility in water [14,15,16,18].

## 3. Materials and Methods

### 3.1. Drug and Excipients

FLL was purchase from Chunxing Pharmacy (Tainan, Taiwan), and OA standard was obtained from Sigma–Aldrich (purity N90%, St. Louis, MO, USA). Analytical grade methanol, ethanol and phosphoric acid were supplied by E. Merck (Darmstadt, Germany). PEG 6000, PEG 4000, dimethicone 100, dimethicone 350, liquid paraffin, Tween 80, Span 60 and Span 80 were obtained from SFL Beauty & Chem. (Tainan, Taiwan). Other chemicals used in the work were all of analytical grade. Pure water was prepared using the Milli-Q system (Millipore, Bedford, MA, USA).

### 3.2. Preparation of FLL Extract

#### 3.2.1. Concoction of FLL Wine Steaming Product

An appropriate amount of FLL was weighed and mixed evenly with Shaoxing wine, followed by soaking for 8 h. Afterward, the mixture was placed in a steamer and steamed for 4 h. FLL was removed when it turned black and was then dried in a 60 °C oven.

#### 3.2.2. Ultrafine Pulverization of FLL

Previous studies [7,8] pointed out that micro-pulverization (1–100 μm) of medicinal materials could disrupt the cell wall and significantly change the amount of leached components. Therefore, FLL was pulverized with a pulverizer and sieved through a 200-mesh sieve before extraction of its components.

#### 3.2.3. Extraction and Concentration of FLL Components

An 85% ethanol was added to the aforementioned FLL powder that underwent ultrafine pulverization for extraction. The ultrafine pulverization powder and solvent ratio was 1:8. The ultrasonicator was set to an extraction temperature of 30 °C and an extraction time of 30 min. After extraction, the solution was filtered and concentrated to a fluid extract using a rotary evaporator. The fluid extract was removed and dried until it became a paste (FLL extract), which was used for the subsequent preparation of dropping pills.

### 3.3. Preparation of FLL Dropping Pills

#### 3.3.1. Preliminary Test

Before using experimental design principles to set the variables and level ranges, screening of the type and range of variables was first carried out to facilitate subsequent evaluation. Hence, we screened the types and ratios of substrates used for preparing dropping pills, the types of condensates, FLL extract loading, and the types and ranges of surfactants. The formulation obtained from the preliminary results were used in a 2^3^ factorial experimental design.

##### Type and Ratio of Substrates

Currently, the most widely used water-soluble substrate for preparing dropping pills is PEG as it is non-toxic to the human body, cheap, and does not easily interact with other components [19,26,31,32]. Hence, water-soluble PEG4000 and PEG6000 were selected as substrates, and the effects of different ratios of PEG 4000:PEG 6000 (5:1, 3:1, 1:1, 1:3, 1:5) on the dropping process of dropping pills were evaluated.

##### Condensate Screening

The choice of condensate is usually made based on the characteristics of the drug and substrates. Usually, oil-soluble condensates used for preparing dropping pills include liquid paraffin and polydimethylsiloxane. Hence, we selected liquid paraffin, polydimethylsiloxane 100, and polydimethylsiloxane 350 as condensates. The effects of these condensates on dropping pill formation were observed and evaluated to select a suitable condensate for subsequent formulation evaluation [33].

##### FLL Extract Loading

Previous studies [34] reported that TCM extracts can affect the formation and quality of dropping pills. Hence, under fixed dropping pill substrate ratios, we added 20, 25, 30, 35, and 40% FLL extract to assess the effects of adding different amounts on dropping pill state and formation.

##### Surfactant Screening

Adding a surfactant to a formulation is a simple and common method to increase the solubility and dissolution rate of a poorly soluble drug [35]. The addition of a surfactant to the dropping pill formulation will not only help in its dissolution but also regulate the hydrophilic–lipophilic balance (HLB) to achieve a reasonable surface tension and aid in adequate dropping pill formation [29]. Hence, in the preliminary test, 5, 10, and 15% of non-ionic surfactants (such as Tween 80, Span 80, and Span 60) were added to assess the effects on dropping pill preparation.

#### 3.3.2. Preparation Process of Dropping Pills

The results of the preliminary test were used to set the variables and level in the 2^3^ factorial experimental design (Table 1). Table 1 lists eight formulations in the 2^3^ factorial experimental design. The total weight of each formulation was 50 g. The hot-melt of a solid dispersion technique was used to prepare dropping pills in this experiment. First, the FLL extract and surfactant were weighed, added to a beaker, and heated using a water bath. Afterwards, different ratios of PEG4000 and PEG6000 were added and melted before mixing well. The mixture was stirred until it was evenly dispersed. Air bubbles were removed, and the mixture was kept at a specific temperature. Next, the mixture was extruded through a 5 mL syringe by gravity, drop by drop slowly into the condensate (such as polydimethylsiloxane 100) at a uniform speed, so the mixture contracts and forms pills that slowly settle to the bottom of the condensation column. After dropping pill condensation was completed, the condensate was filtered away, and the dropping pills were collected. The dropping pills were dried naturally at room temperature for subsequent experiments [28,29]. The schematic diagram of FLL dropping pill preparation showed in Figure 13.

#### 3.3.3. The 2^3^ Experimental Design 

A 2^3^ factorial experimental design was used to set the formulation factors for preparing dropping pills based on the variables and level ranges obtained from the preliminary test screening to obtain eight experimental groups [28]. As shown in Table 1, three formulation variables, the ratio of PEG4000:PEG6000 (A), FLL extract loading (B), and percentage of Tween 80 (C) were selected. Two levels (low level and high level) were set for each factor to assess their effects on dropping pill roundness, weight variation, and disintegration time, and the optimal dropping pill formulation was selected for the dissolution experiment. The optimal formulation conditions were set as roundness close to 1, weight variation <5%, and a disintegration time of 5–15 min. A commercially available statistical software package (DESIGN EXPERT V 6.0.3, Minneapolis) was used to evaluate of the results by analysis of variance (ANOVA).

### 3.4. Physical Evaluation of FLL Dropping Pills 

#### 3.4.1. Roundness

Twenty dropping pills were randomly selected from each batch, and a digital camera was used to take photographs. The scale bar zoom function was used to calculate the axial and equatorial diameters of the dropping pills. Sphericity was assessed and presented as elongation. The calculation formula for elongation is as follows. The closer the calculated value was to 1, the pill was considered as round [29,33].
Elongation = axial diameter/equatorial diameter

#### 3.4.2. Weight Variation 

Twenty dropping pills were randomly selected from each batch, and the weights of individual pills were precisely measured. The mean weight was calculated. The calculation formula is as follows:Weight variation = (standard deviation of weight/mean weight) × 100%

#### 3.4.3. Disintegration Test

Disintegration time was determined according to Pharmacopoeia of the People’s Republic of China [36]. A measure of 1.2 g of dropping pills was packed in No. 0 capsule shells and placed in a glass tube in the disintegration analyzer (Hsiangtai Machinery Industry, New Taipei City, Taiwan) with 1000 mL of pure water, which was maintained at 37 ± 1 °C. Six dropping pills of each formulation were tested.

### 3.5. Evaluation of the Optimal FLL Dropping Pill Formulation

The aforementioned 2^3^ factorial experimental design was used to screen for the optimal formulation in the following analyses.

#### 3.5.1. Solubility Study

The optimal FLL dropping pills were pulverized and placed in R.O. water. The samples were shaken at 50 rpm for 24 h at 37 ± 0.5 °C, then the supernatant was collected and filtered through 0.45 μm syringe filters (Millex-GV, Millipore, Bedford, MA, USA) for OA content analysis using HPLC.

#### 3.5.2. Content Uniformity

A measure of 1.2 g of FLL dropping pills (*n* = 10) was precisely weighed and pulverized. The powder was added to a volumetric flask, and ethanol was added for analysis. Next, HPLC was used to measure the OA content, and the C.V.% was calculated.

#### 3.5.3. HPLC Analysis Conditions for OA

Reverse phase chromatography was used for this experiment. The chromatography column was an Inertsil ODS-3 (5 μm, 250 mm × 4.6 mm), the column temperature was set to 30 °C, the mobile phase was methanol: 0.1% phosphoric acid aqueous solution (90:10), the flow rate was set to 0.6 mL/min, and the detection wavelength was 210 nm.

#### 3.5.4. Dissolution Test 

The optimal formulation of FLL dropping pills and the commercially available FLL medicines were used for the dissolution experiment. The basket method described in the Appendix of the 8th Chinese Pharmacopoeia was used for the experiment [37]. The dissolution media consisted of 500 mL 0.5% sodium dodecyl sulfate solution, and the rotation speed was set 100 rpm. The temperature was set to 37 °C ± 0.5 °C. The sampling time points were 5, 10, 15, 30, 45, 60, min, and 120 min. At each time point, 2 mL of sample was collected and passed through a 0.22 μm filter before HPLC was used to analyze the amount of drug released.

## 4. Conclusions

This study successfully employed the solid dispersion technique to produce FLL dropping pills. In addition to increasing the water solubility of FLL, we also enhanced its application value and improved the issues of current dosage administration forms. We hope that these results and concepts can be providing as a reference for the development of dosage forms in the future.

## Figures and Tables

**Figure 1 pharmaceuticals-15-01433-f001:**
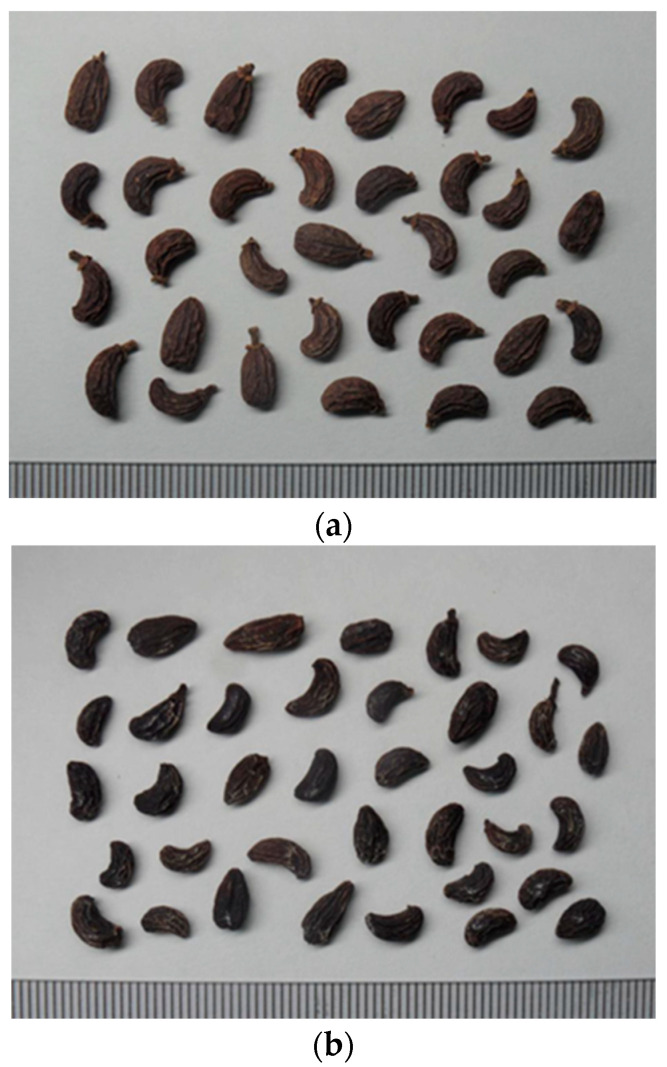
Comparison of FLL photographs before (**a**) and after (**b**) concoction.

**Figure 2 pharmaceuticals-15-01433-f002:**
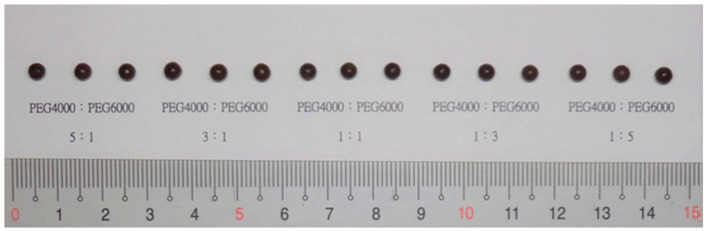
Dropping pills prepared using different ratios of PEG4000 and PEG6000.

**Figure 3 pharmaceuticals-15-01433-f003:**
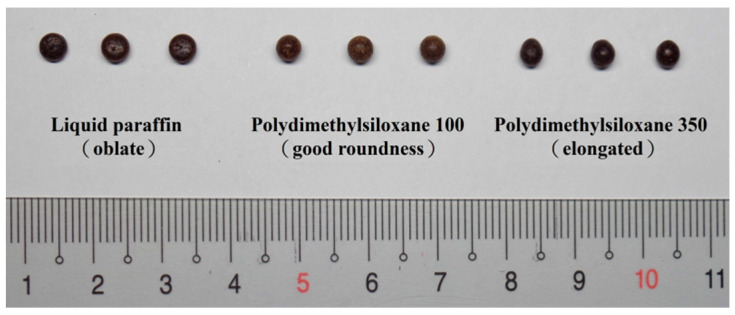
Dropping pills prepared using different types of condensate.

**Figure 4 pharmaceuticals-15-01433-f004:**
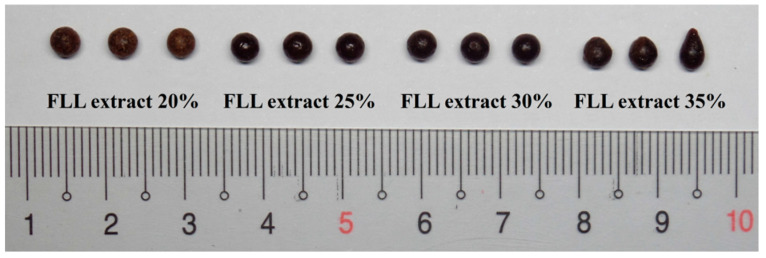
Dropping pills prepared using different amounts of FLL extract.

**Figure 5 pharmaceuticals-15-01433-f005:**
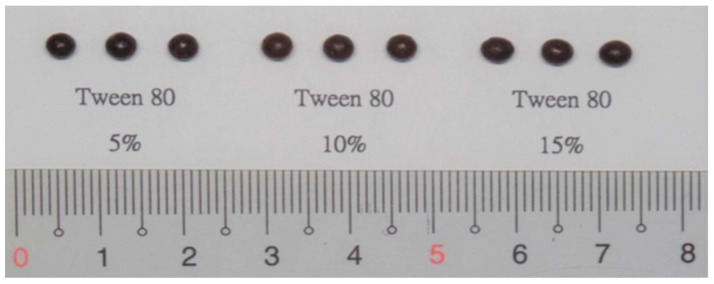
Dropping pills prepared by adding different concentrations of Tween 80.

**Figure 6 pharmaceuticals-15-01433-f006:**
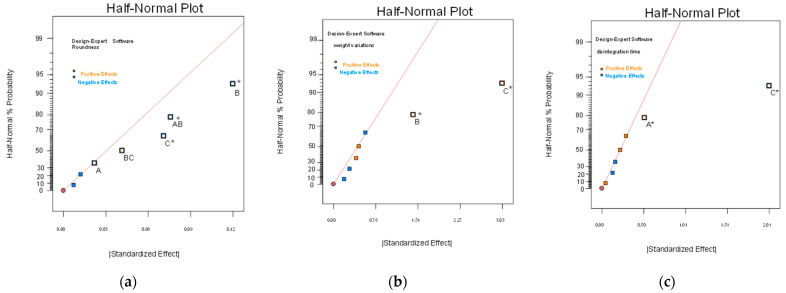
The half-normal plots for identifying significant effects (* *p* < 0.05). (**a**) The effect on roundness, (**b**) the effect on weight variation, and (**c**) the effect on the disintegration time. The ratio of PEG 4000/PEG 6000 (A), FLL extract loading (B), percentage of Tween 80 (C).

**Figure 7 pharmaceuticals-15-01433-f007:**
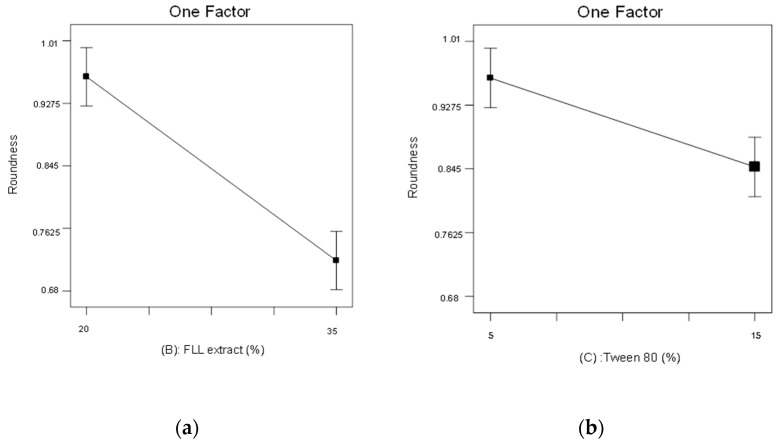
Using the 2^3^ factorial experimental design to evaluate the significant effects of factors B and C on roundness (*p* < 0.05). (**a**) factor B: FLL extract loading, (**b**) factor C: Tween 80 content.

**Figure 8 pharmaceuticals-15-01433-f008:**
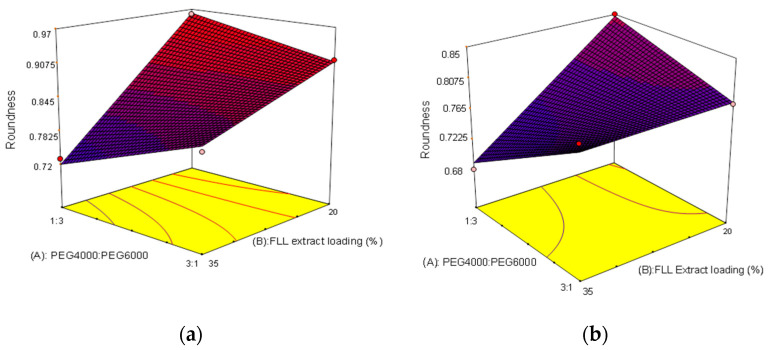
Using a 2^3^ factorial experimental design to evaluate the significant effects of the interaction between ratio of PEG 4000/PEG 6000 (A) and FLL extract loading (B) on roundness under different Tween 80 concentrations (*p* < 0.05). (**a**) 5% Tween 80 (**b**) 15% Tween 80.

**Figure 9 pharmaceuticals-15-01433-f009:**
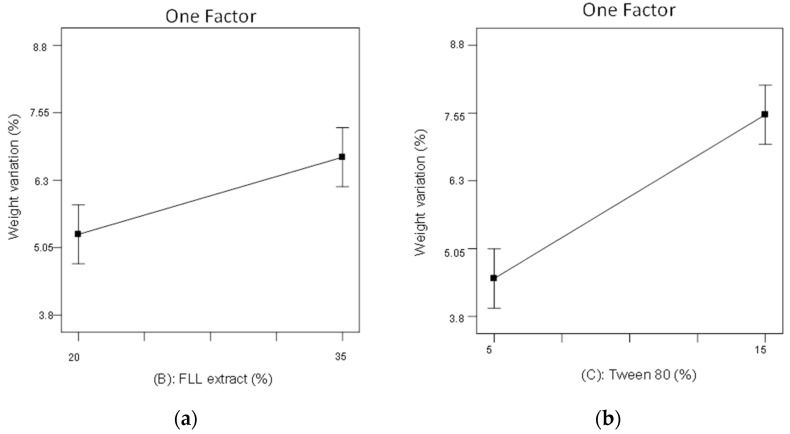
Using a 2^3^ factorial experimental design to evaluate the effects of significant variables on weight variation (*p* < 0.05). (**a**) FLL extract loading (factor B). (**b**) percentage of Tween 80 (factor C).

**Figure 10 pharmaceuticals-15-01433-f010:**
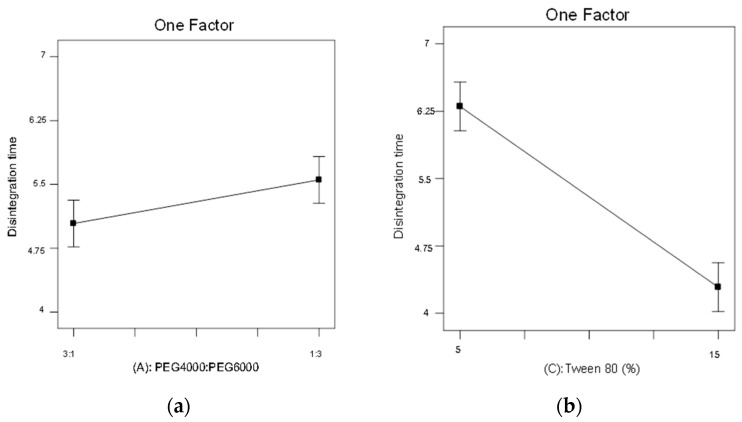
Using a 2^3^ factorial experimental design to evaluate the effects of significant variables on disintegration time (*p* < 0.05). (**a**) PEG 4000: PEG 6000 (factor A), (**b**) percentage of Tween 80 (factor C).

**Figure 11 pharmaceuticals-15-01433-f011:**
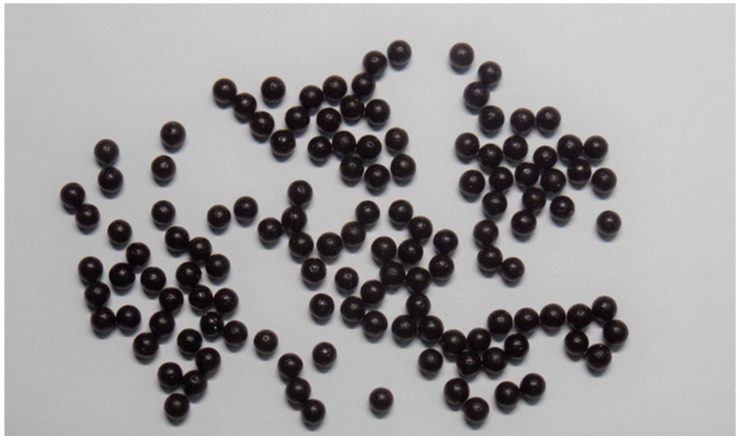
Appearance of the optimal formulation dropping pill.

**Figure 12 pharmaceuticals-15-01433-f012:**
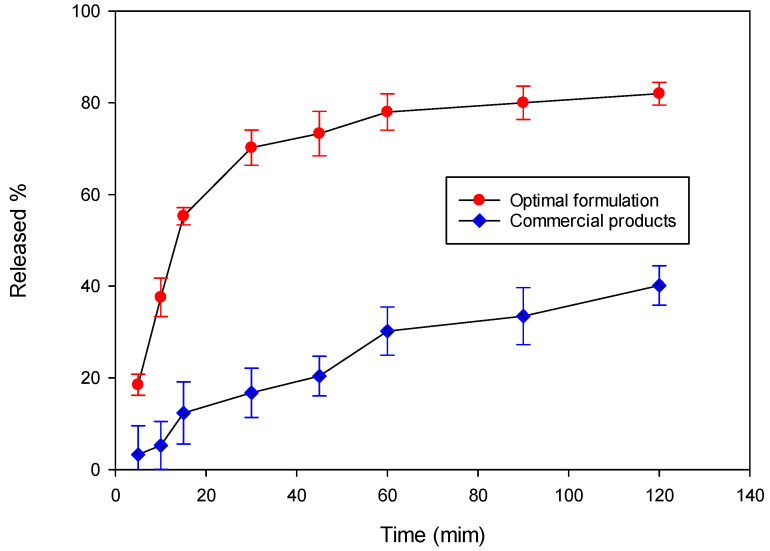
Dissolution curves of the optimal formulation FLL dropping pill and commercial Chinese patent medicines.

**Figure 13 pharmaceuticals-15-01433-f013:**
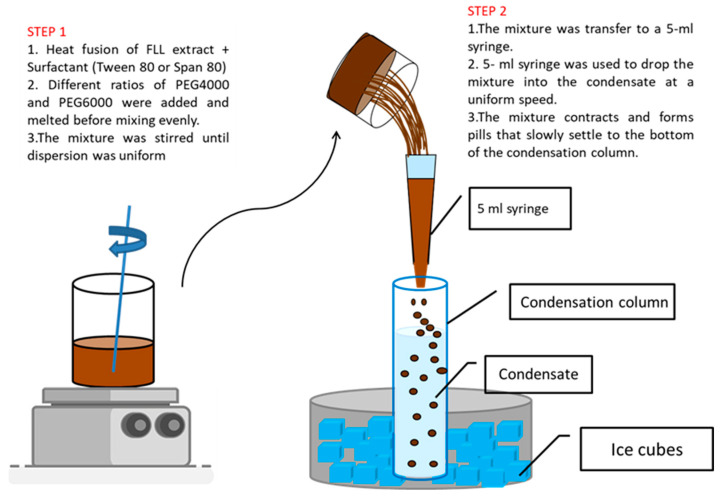
Schematic diagram of FLL dropping pill preparation.

**Table 1 pharmaceuticals-15-01433-t001:** The 2^3^ full factorial design matrix and response results for dropping pills.

**I** **ndependent Variables**				**Dependent Variable**		
**Formulation**	**A**	**B**	**C**	**Roundness**	**Weight Variation(%** **)**	**Disintegration Time** **(min)**
F1	1:3	35	15	0.68 ± 0.06	8.14	4.68 ± 0.20
F2	3:1	35	5	0.81 ± 0.04	4.54	6.09 ± 0.16
F3	3:1	20	5	0.91 ± 0.05	4.17	5.84 ± 0.29
F4	1:3	35	5	0.73 ± 0.05	5.43	6.94 ± 0.23
F5	3:1	35	15	0.81 ± 0.04	8.79	4.06 ± 0.36
F6	1:3	20	5	0.96 ± 0.02	3.85	6.34 ± 0.30
F7	3:1	20	15	0.79 ± 0.04	7.12	4.17 ± 0.12
F8	1:3	20	15	0.85 ± 0.03	6.04	4.25 ± 0.17
**Independent variables**			**Levels**			
			**Low level**	**High level**		
Ratio of PEG4000/PEG6000 (A) ^c^			3:1	1:3		
FLL extract loading (B) ^a,b^			20%	35%		
Percentage of Tween 80 (C) ^a,b,c^			5%	15%		

^a^ The independent variables have a significant effect on roundness (*p* < 0.05). ^b^ The independent variable has a significant effect on weight variation (*p* < 0.05). ^c^ The independent variable has a significant effect on disintegration time (*p* < 0.05).

**Table 2 pharmaceuticals-15-01433-t002:** The results of content uniformity for optimal dropping pill formulation (*n* = 10).

No.	Content Uniformity (%)
1	92.69
2	88.32
3	94.42
4	93.40
5	92.08
6	89.34
7	96.45
8	93.10
9	89.34
10	86.29
Mean(%)	91.54
S.D.	3.11
C.V.%	3.40

**Table 3 pharmaceuticals-15-01433-t003:** The results of solubility for optimal dropping pill formulation (*n* = 6).

	Mean (μg/mL)	S.D.	C.V.%
Optimal formulation	70.60	2.40	3.40

## Data Availability

Data is contained within the article.

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
