# Peer review of "Optimization of Formulation Parameters in Preparation of Fructus ligustri lucidi Dropping Pills by Solid Dispersion Using 23 Full Experimental Design"

_pharmaceuticals, 2022, doi:10.3390/ph15111433_

Round 1

Reviewer 1 Report

The manuscript presents a very interesting subject and innovative approach to design a new presentation for a traditional Chinese medicine ingredient. Nevertheless, the whole study is still a work in progress, where statistics are shown raw. Nowadays researchers trust that statistical analysis is being done rigorously as it is required; therefore, ANOVA tables are seldom needed in figures for graphical presentation. More likely, the stat info is included as part of the figures or figure’s legend.

The results are discussed briefly and little awkward. Factors A, B, C, and their interactions should be addressed in the text by the corresponding parameter they represent. Example, instead of factor A use PEG ratios, and Extract loading value instead of B. In this manner, the audience will easily understand the factors’ role in the parameter discussed, like roundness.

In lines 472 to 477. For roundness the term sphericity is suggested instead; and diameter measurements could be axial and equatorial, instead of short and long.

Please drop the ANOVA tables and stat software graph. Keep the surface analysis graphs. A table of content uniformity and a graph on the dissolution test data would be most welcome. For instances, for Table 1, elaborate a table with independent variables on the right column and the average plus minus standard deviations values for parameters like weights to the left. Those values of interest should be highlighted, and their significance addressed in the figure/table legend.

By the end of discussion, a brief explanation of why the formulation regarded as optimal is recommended. The twin content is not presented here as a factor for better solubility of FLL and should be. As stated before, this is a work in progress and I strongly advice the authors to resubmit their work as soon as graph and new tables are included.

Author Response

Dear reviewer:

Thank you very much for providing us the valuable comments. The comments were valuable and helpful for improving our paper. According to your comments and suggestions, we have revised the manuscript and responded, point by point, to the comments, please see the attachment.

Kind regards,   Yuh-Tyng Huang Ph.D

Reviewer 2 Report

There is a good rationale and investigation design for the development of dropping pills. 

  1. Authors must correct the numerous typographical errors in the manuscript. 
  2. The scientific name of the plant must be written in italics
  3. It is unclear why powder and decoction dosage forms are ‘prohibitive administration modes’; the author must clarify the challenge it presents to the administration of FLL. 
  4. The quantitative solubility data (e.g., g/mL) of FLL or OA must be presented to draw a good basis for the proposed preparation technique. The ‘modes’ that the authors try to present is confusing; it is assumed that it is also an oral route, hence, I cannot judge if the current form will result in a better route or mode of delivery. 
  5. Authors must describe briefly why it is called ‘dropping pills’
  6. Authors must describe precisely the role of the syringe in extruding or forming the pills prepared. The term ‘condensate’ in line 447 appears to be incorrect. Do you mean condenser setup? The setup which holds the formed ‘dropping pills’ should be described too. 

Author Response

(The authors gave the same response as above.)

Round 2

Reviewer 1 Report

The authors included all suggestions and the manuscript has improved very much